# Revalorization of Cava (Spanish Sparkling Wine) Lees on Sourdough Fermentation

Alba Martín-Garcia [1,2,3], Montserrat Riu-Aumatell [1,2,3,*] and Elvira López-Tamames [1,2,3]

1   Departament de Nutrició, Ciències de l'Alimentació i Gastronomia, Facultat de Farmàcia i Ciències de l'Alimentació, Campus de l'Alimentació de Torribera, Universitat de Barcelona, Av. Prat de la Riba 171, 08921 Santa Coloma de Gramenet, Spain; albamartin@ub.edu (A.M.-G.); e.lopez.tamames@ub.edu (E.L.-T.)

2   Institut de Recerca en Nutrició i Seguretat Alimentària (INSA·UB), Universitat de Barcelona, Av. Prat de la Riba 171, 08921 Santa Coloma de Gramenet, Spain

3   Xarxa d'Innovació Alimentària de la Generalitat de Catalunya (XIA), C/Baldiri Reixac 4, 08028 Barcelona, Spain

\*   Correspondence: montseriu@ub.edu; Tel.: +34-93-4033795

**Abstract:** Cava lees are a sparkling wine by-product formed of dead microorganisms, tartaric acid and other inorganic compounds, with a potential for enhancing microbial growth. Lees are rich in antioxidant compounds as well as β-glucans and mannoproteins. The aim of this study was to evaluate the effect of different concentrations of cava lees (0–2% *w*/*w*) on the microbiota (LAB and yeasts) responsible for sourdough fermentation (8 days) to revalorize this by-product of the wine industry. The results showed that 2% cava lees promoted microbial growth and survival in both wheat and rye sourdoughs, except for yeast growth in rye, which stopped at day 3 of fermentation. Moreover, sourdough with lees achieved lower pH values as well as higher concentrations of organic acids, especially lactic and acetic acids ($p < 0.05$). To sum up, the use of cava lees in sourdough formulation promotes the growth and survival of microorganisms, which, in consequence, promotes a lower pH and greater amounts of organic acids. This could lead to microbial stability as well as changes in bread flavor.

**Keywords:** cava lees; wine by-product; revalorization; sourdough; lactic acid bacteria; yeasts; prebiotic

## 1. Introduction

Cava is a Spanish sparkling wine with a Certified Brand of Origin (CBO) that is produced using a traditional method, refermenting a base wine in a sealed bottle. In order to be considered Cava, wines must undergo an ageing process for a minimum of 9 months (EC Regulation 2019/934). Yeast autolysis takes place during the ageing process, releasing cell components and breakdown products into the wine [1,2].

Lees are defined as the residue formed at the bottom of receptacles containing wine, after fermentation and during storage (e.g., during the ageing process of Cava). Lees mostly consist of dead microorganisms (generally *Saccharomyces cerevisiae*), tartaric acid and other adsorbed compounds [3]. The cell wall of *S. cerevisiae* remains intact and is mainly composed of mannoproteins and branched β-glucans, as well as soluble polysaccharides [1]. Cava lees are rich in antioxidant compounds [3,4] along with soluble and insoluble dietary fiber from the yeast cell wall [5].

Each bottle of Cava contains about 1g of lees, which contains approximately $10^8$ yeast cells that contribute to organoleptic properties during ageing [6]. However, yeast lees of Cava are considered a by-product of approximately 300 tons per year, representing approximately 25% of the waste by-products from the wine industry [4]. Despite being the second largest by-product in wineries, wine lees are mainly destined for distillation. However, considering their composition, those lees could also potentially acquire an added

value [5,7–9]. In fact, there is an increasing trend in the food and drinks industry to reduce food waste by revalorizing by-products and co-products, therefore contributing to a circular economy and more sustainable food production [4,8,10–16].

On the other hand, consumers are more conscious of their food consumption and their health. Moreover, artisan food products' popularity is increasing due to a clean-label trend, including the use of sourdough in bread-making [17]. In addition, Spain has recently developed new bread legislation, including the definition of sourdough and establishing some rules regarding its production and labelling (RD 308/2019) [18].

Sourdough is a mixture of flour and water, fermented by homo- and heterofermentative lactic acid bacteria (LAB) and yeasts. The traditional method consists of the spontaneous fermentation of sourdough by the microorganisms present in the flour, which are responsible for acidification, leavening and flavor formation [17,19,20]. Different flours (e.g., wheat, rye, teff, barley, etc.) may be used to produce sourdough, presenting different characteristics in the final bread, such as flavor or nutritional value. In fact, the flour composition and its quality can affect the microorganisms' dynamics and, therefore, the sensory properties of the sourdough bread [17,21–23].

Therefore, the aim of this study was to evaluate the prebiotic effect of yeast lees on the microbiota (LAB and yeasts) responsible for sourdough fermentation to revalorize this by-product of the wine industry.

## 2. Materials and Methods

### 2.1. Preparation and Propagation of Sourdoughs

A commercial wheat flour was used for sourdough preparation (Ref.: 7230 Buonpane, Molino Quaglia SpA, Padua, Italy), with the following composition (% *w/w*): carbohydrates 72.0, fat 1.5, fiber 2.0, protein 11.5 and moisture 15.0. A second type of sourdough was prepared using a commercial rye flour (Ref.: 50782, Molino Quaglia SpA, Padua, Italy), with the following composition (% *w/w*): carbohydrates 76.4, fat 0.8, fiber 4.6, protein 4.6 and moisture 15.0.

Both types of sourdough were prepared by mixing 100 g of flour and 100 mL of sterile distilled water, without the inoculation of starter culture bacteria or yeasts, and incubated at room temperature. Cava lees were provided by the winery Freixenet S.A. (Sant Sadurní d'Anoia, Spain) and lyophilized following the method described by Hernández-Macias et al., (2021) [7]. Lyophilized lees were added at different concentrations (0%, 0.5%, 1% and 2% (*w/w*)) to assess their effect on sourdough fermentation (Table 1). The sourdoughs were propagated daily by backslopping for 8 days, inoculating an aliquot of the previous dough into a new mixture of flour and water. All fermentations were carried out in triplicate.

**Table 1.** Ingredients of sourdough (flour weight basis, g).

|  | Flour [1] | Water | Dough [2] | Cava Lees [3] |
|---|---|---|---|---|
| Control | 100 | 100 | 100 | - |
| 0.5% Lees | 100 | 100 | 100 | 0.5 |
| 1% Lees | 100 | 100 | 100 | 1 |
| 2% Lees | 100 | 100 | 100 | 2 |

[1] Either wheat or rye flour; [2] Aliquot of the previous dough into the new mixture; [3] Lees were added as a percentage of flour weight in sourdough formulation in each propagation step.

### 2.2. Viable Counts of Lactic Acid Bacteria (LAB) and Yeasts

To assess the microbial growth of LAB and yeasts, samples of 10 g of sourdough were added to 90 mL of sterile peptone water (Ref.: 1402, Condalab, Madrid, Spain) and homogenized with a laboratory blender (Stomacher 400 Seward Ltd., Worthing, UK) for 1 min. Samples were taken daily, diluted and plated in MRS (Ref.: 1043, Condalab, Madrid, Spain) to monitor LAB populations and in Saboraud-Chloramphenicol Agar (Ref.: 01-166-500; Scharlab, Barcelona, Spain) for yeasts.

### 2.3. Determination of pH, Fermentation Quotient (FQ) and Organic Acids

Sourdough fermentation was monitored daily by pH using the pH meter XS PH60 VioLab (XS Instruments, Carpi, Italy). The fermentation quotient (FQ) was determined as the molar ratio between lactic and acetic acids. Organic acids (acetic, citric, D-lactic, L-lactic and L-malic acids) were determined using enzymatic detection kits supplied by BioSystems (Barcelona, Spain) and a spectrophotometer Shimadzu UV-3600 (Shimadzu Corporation, Kyoto, Japan), following each kit's instructions.

### 2.4. Statistical Analysis

All assays were performed in triplicate. Statistical analysis was performed using the Prism 9 v.9.1.2 (225) (GraphPad Software, LLC., San Diego, CA, USA) statistical package. The results are reported as means ± standard error (SE) for parametric data. Analysis of variance (ANOVA) and comparison of the means were conducted using Tukey's test, with a confidence interval of 95% and significant results with a *p*-value of $\leq 0.05$. Principal component analysis (PCA) was also performed to determine the differences between sourdoughs.

## 3. Results and Discussion

Two types of flour (wheat and rye) were used to produce sourdoughs, and different concentrations of yeast lees (0%, 0.5%, 1% and 2% (*w/w*)) were added to the sourdough formulation to test its prebiotic effect on the fermenting microbiota.

### 3.1. Propagation of Sourdoughs

3.1.1. Effect of Cava Lees on Lactic Acid Bacteria

In both wheat and rye sourdoughs, adding 2% of cava lees resulted in major viable LAB cells (8.4 ± 0.2 log CFU/mL and 9.1 ± 0.1 log CFU/mL, respectively) at the end of fermentation (Figure 1). These results are in accordance with Hernández-Macias et al. (2021) [7], who reported higher microbial counts in vitro with 2% and 5% of yeast lees after 24h and 48h of incubation, respectively.

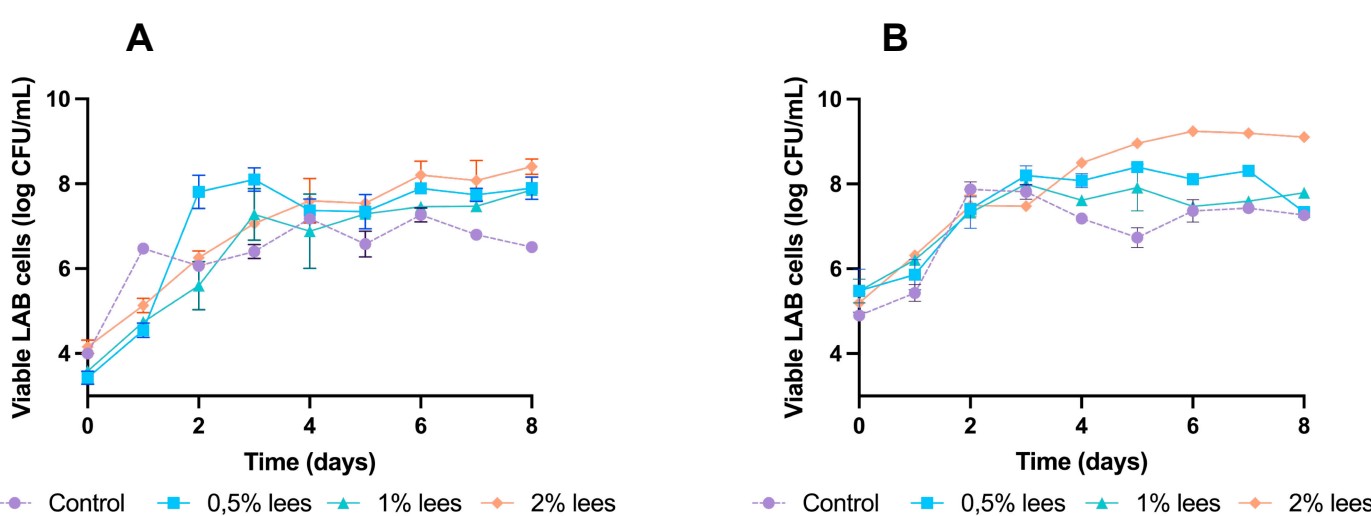

**Figure 1.** Growth kinetics of LAB during sourdough fermentations: (**A**) wheat sourdoughs; (**B**) rye sourdoughs.

As shown in Figure 1A, the maximum growth achieved by LAB in wheat sourdough was: 7.3 ± 0.2 log CFU/mL (day 6) in control fermentations; 8.1 ± 0.3 log CFU/mL (day 3) by adding 0.5% lees; 7.9 ± 0.1 log CFU/mL (day 8) by adding 1% lees; and 8.4 ± 0.2 log CFU/mL (day 8) by adding 2% lees.

On the other hand, Figure 1B shows the growth kinetics of rye sourdoughs, reaching their highest number of viable cells as follows: 7.9 ± 0.2 log CFU/mL (day 2) in control

fermentations; $8.4 \pm 0.1$ log CFU/mL (day 5) with 0.5% lees; $8.0 \pm 0.1$ log CFU/mL (day 3) with 1% lees; and $9.3 \pm 0.1$ log CFU/mL (day 6) with 2% lees.

These results suggest that the incorporation of Cava lees in the formulation of wheat and, in particular, rye sourdough improve the growth and survival of LAB. In wheat sourdough (Figure 1A), it can be observed that bacterial growth increases with lees concentration, obtaining the best results at 2% (*w*/*w*), and having statistically significant differences in all fermentations with lees regarding control.

In rye sourdough (Figure 1B), at the end of fermentation, incorporating lees in its formulation has an effect with 1% (*w*/*w*), obtaining statistically significant results with 2% (*w*/*w*) of Cava lees.

Other studies have also reported a stimulatory effect on fermenting LAB by different by-products [7,15,24–26], mainly due to oligo- and poly-saccharides. As stated by Rivas et al. (2021) [5], wine lees are the winery by-product with the highest percentage of dietary fiber (DF), over grape skins and stems. Similarly, the positive effect that Cava lees had over LAB's growth could be attributed to the use of the β-glucans and mannoproteins found in their cell wall as a carbon source. Moreover, several studies focused on the extraction and usage of β-glucans and mannoproteins from various sources (spent brewer yeasts, bacterial production or cereal origin) in order to use them as food ingredients [10,15,26–28].

### 3.1.2. Effect of Cava Lees on Yeasts

Figure 2 shows the growth development of yeasts in both wheat (Figure 2A) and rye (Figure 2B) sourdoughs. It can be observed that, in wheat control sourdoughs, the plate counts showed no yeast growth.

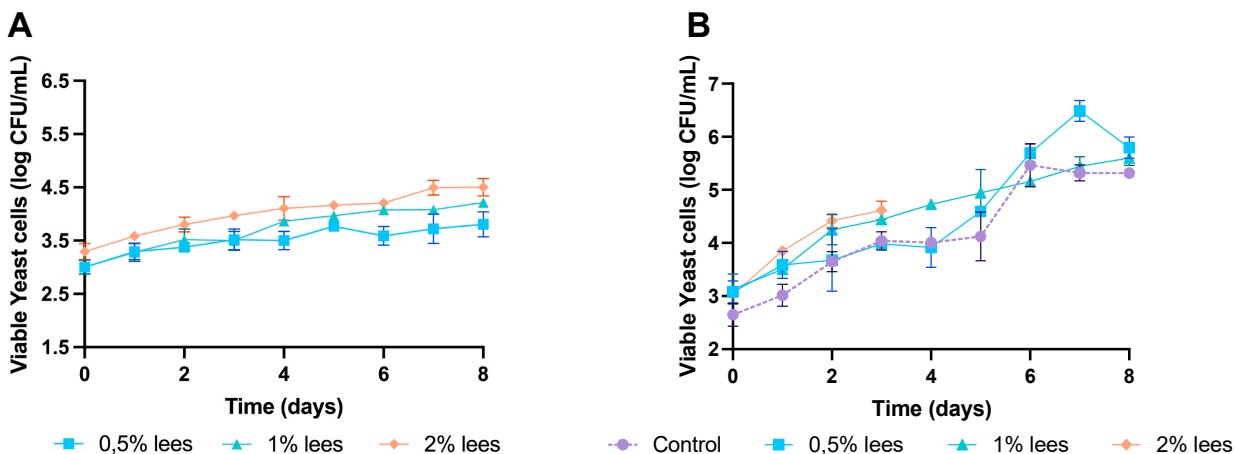

**Figure 2.** Growth kinetics of yeasts during sourdough fermentations: (**A**) wheat sourdoughs; (**B**) rye sourdoughs.

Regarding yeasts in wheat sourdough (Figure 3), cell viability slightly increased with lees ($p > 0.05$), being stable during the whole fermentation process, and obtained the highest number of viable cells at the end of fermentation: $3.8 \pm 0.2$ log CFU/mL with 0.5% lees and $4.2 \pm 0.1$ log CFU/mL with 1% lees; and $4.5 \pm 0.2$ log CFU/mL with 2% lees.

In contrast, rye sourdough (Figure 3) control samples reached a yeast cell density of $5.3 \pm 0.1$ log CFU/mL at the end of fermentation, whereas the samples with 2% Cava lees stopped yeast growth at day 3. The sourdough environment is considered to be stressful; consequently, the microbiota has to adapt to the variability in nutrients and the low pH [17,29]. On that account, wheat control sourdoughs had the fastest acidification (data not shown), which may explain the growth inhibition of yeasts. Yeast higher plate counts in rye sourdoughs were as follows: $5.5 \pm 0.4$ log CFU/mL (day 6) in control fermentations; $6.5 \pm 0.2$ log CFU/mL (day 7) with 0.5% lees; and $5.6 \pm 0.1$ log CFU/mL

(day 8) with 1% lees. In fact, rye sourdoughs presented higher populations of yeast from the beginning than wheat (data not shown).

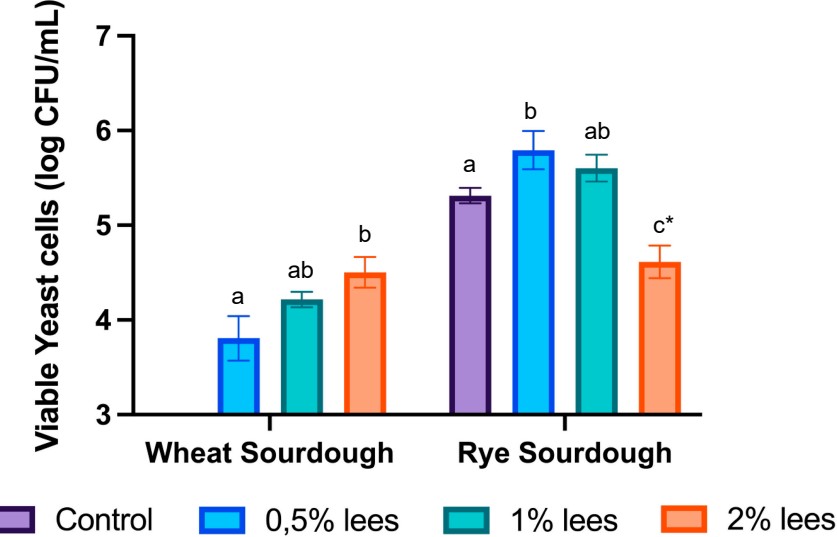

**Figure 3.** Yeast cell density at the end of sourdough fermentation. Different letters denote statistically significant differences ($p < 0.05$) between the sourdoughs with different amounts of lees. * Yeast plate counts in rye sourdough with 2% of cava lees ended at day 3 of fermentation, i.e., there was no further growth.

Similarly to LAB, higher yeast cell densities may be due to the fiber composition of Cava lees. Lees' main monosaccharides are glucose, mannose and rhamnose [5], whereas the carbohydrates present in the flour are sucrose, glucose, fructose and maltose [30]. Therefore, lees present an additional source of glucose that yeasts can catabolize.

In both wheat and rye sourdoughs, it took approximately 6 days to obtain a stable microbiota, which is in accordance with other studies [31,32]. The sourdough ecosystem is formed by LAB and yeasts that can interact with each other. These interactions can be synergistic (positive effects) or antagonistic (negative effects). As a result, both LAB and yeast may improve their growth kinetics or decrease them [17,30,32]. For instance, the absence of yeast growth in wheat control fermentations may be due to an antagonistic interaction with LAB, such as acidification of the medium or the production of some antifungal compounds [33].

On the other hand, in sourdough, it is very common that the association between LAB and yeast is a consequence of their metabolism preferences (e.g., a maltose-positive LAB with a maltose-negative yeast). For instance, sucrose is hydrolyzed by yeasts, releasing glucose and fructose for LAB to consume [34]. The extra nutrients from lees could enhance that type of microbial association.

Overall, the addition of Cava lees into the formulation of sourdough at 2% ($w/w$) improved the growth and survival of the dough microorganisms.

### 3.2. Physicochemical Characterization of Sourdoughs

In addition to microbial growth, pH was also monitored daily. Furthermore, at the beginning and end of fermentation, several organic acids (acetic, citric, lactic and malic acids) were analyzed, and, consequently, the fermentation quotient (FQ) was determined for both wheat and rye sourdoughs. FQ is a molar ratio between the values of lactic and acetic acids.

### 3.2.1. Monitoring of pH

New Spanish legislation (RD 308/2019) [18] establishes that sourdough must have a maximum pH of 4.2 before incorporation into the bread. Following those guidelines, wheat

control sourdoughs have a slightly higher final pH, with a value of 4.38 ± 0.04. On the contrary, sourdoughs including Cava lees have an acidic pH, with a difference of 0.50 relative to control. In fact, 1% (*w/w*) fermentations have a lower value of 3.87 ± 0.13, followed by 2% (*w/w*) sourdoughs (3.88 ± 0.04) and, finally, 0.5% (*w/w*) samples (3.98 ± 0.04). This tendency in pH reduction is in accordance with the higher LAB cell densities that increase with the addition of lees.

Conversely, all rye sourdoughs meet legislation requirements, obtaining a lower pH with 2% (*w/w*) Cava lees (3.66 ± 0.01), also relating to higher bacterial populations. Moreover, the addition of lees significantly decreased the initial pH with a difference of 0.66 regarding control.

Overall, both wheat and rye sourdoughs showed a reduction in pH (ranging between 1.74 and 1.95 in wheat; and 1.97 and 2.10 in rye) with the addition of Cava lees, in accordance with other studies that included Cava lees [7,8], co-products [12] or by-products [25,35] from other food industries with potential revalorization. In all cases, the ingredient added to the formulation was already acidic (e.g., citrus or orange fibers), similar to the Cava lees, which are also acidic.

### 3.2.2. Fermentation Quotient (FQ) and Organic Acids

Regarding organic acids (Table 2), the addition of lees to sourdough formulation resulted in significantly higher concentrations of organic acids, in both the beginning and end of fermentation in both wheat and rye formulations, following the same tendency as reported by Vriesekoop et al. (2021) [15], with the addition of brewer spent grain to sourdough bread production. In contrast, the FQ decreased with the addition of lees, with a difference in control of 6.6 in wheat (50% lower) and 3.39 in rye (52% lower), with 2% of Cava lees. In fact, the FQ was lower in all rye samples compared to wheat values. This is the result of the lower production of lactic acid and the higher concentrations of acetic acids in rye sourdoughs [31]. In fact, high values of FQ are usually found in traditional sourdoughs [30,31,36] and could be attributed to a larger presence of homofermentative and facultative heterofermentative LAB, which primarily converts glucose into lactic acid, with respect to obligate heterofermentative LAB, which also produce acetic acid [17,30].

Wheat sourdoughs (Table 2) showed an increase in initial organic acids (L-malic and citric acid) when Cava lees were added to the formulation. L-malic acid increased significantly from 0.450 ± 0.070 g/L in the control to 0.850 ± 0.021 g/L in 0.5% sourdoughs, reaching 1.000 ± 0.045 g/L in 2% sourdoughs. The concentration of citric acid ranged between 22.840 ± 2.418 mg/L (control) and 359.250 ± 11.341 mg/L (2% Cava lees). Citric acid increased significantly with the addition of at least 1% of Cava lees to the formulation (174.172 ± 12.815 mg/L), having a major impact with 2% lees.

At the end of fermentation (8 days), the use of Cava lees augmented the concentration of the quantified organic acids, having an effect at 1% and, in particular, in 2% lees sourdoughs. The production of acetic acid was five times greater in 2% cava lees sourdoughs (0.660 ± 0.023 g/L) than in the control (0.135 ± 0.025 g/L). Lactic acid concentration increased 2.5 times in 2% fermentations in comparison to the control. The higher production of organic acids is in accordance with a higher cell density and a lower pH with the addition of Cava lees.

As for FQ, there was a significant decrease in control fermentations when a minimum of 0.5% (*w/w*) Cava lees were added to the sourdough, although there were no differences between 1% and 2% sourdoughs.

Organic acids in rye sourdoughs (Table 2) also increased by adding Cava lees to their formulations. In comparison to wheat, L-malic and citric acid concentrations were higher in the control already and were critically augmented with the addition of lees.

**Table 2.** Physicochemical characterization of sourdoughs (both wheat and rye) and fermentation quotient (FQ).

| Wheat Sourdoughs | Control | 0.5% lees | 1% lees | 2% lees |
|---|---|---|---|---|
| Beginning of fermentation (t = 0 days) | | | | |
| pH | 5.78 ± 0.04 | 5.72 ± 0.07 | 5.48 ± 0.03 | 5.83 ± 0.04 |
| Acetic acid (g/L) | <0.03 | <0.03 | <0.03 | <0.03 |
| Lactic acid (g/L) | <0.02 | <0.02 | <0.02 | <0.02 |
| Malic acid (g/L) | 0.45 ± 0.07 [a] | 0.85 ± 0.02 [b] | 0.95 ± 0.04 [b] | 1.00 ± 0.05 [b] |
| Citric acid (mg/L) | 22.84 ± 2.42 [a] | 40.74 ± 8.70 [a] | 174.17 ± 12.82 [b] | 359.25 ± 11.34 [c] |
| End of fermentation (t = 8 days) | | | | |
| pH | 4.38 ± 0.05 [a] | 3.98 ± 0.04 [b] | 3.87 ± 0.13 [b] | 3.88 ± 0.04 [b] |
| Acetic acid (g/L) | 0.14 ± 0.03 [a] | 0.29 ± 0.08 [a] | 0.51 ± 0.02 [b] | 0.66 ± 0.023 [b] |
| Lactic acid (g/L) | 2.58 ± 0.14 [a] | 3.14 ± 0.34 [a] | 4.37 ± 0.13 [b] | 6.35 ± 0.25 [c] |
| Malic acid (g/L) | <0.03 | <0.03 | <0.03 | <0.03 |
| Citric acid (mg/L) | <11.00 | <11.00 | <11.00 | <11.00 |
| *Fermentation Quotient (FQ)* | 13.00 ± 0.96 [a] | 7.40 ± 0.89 [b] | 6.26 ± 0.44 [b] | 6.40 ± 0.02 [b] |
| **Rye Sourdoughs** | **Control** | **0.5% lees** | **1% lees** | **2% lees** |
| Beginning of fermentation (t = 0 days) | | | | |
| pH | 6.29 ± 0.07 [a] | 6.01 ± 0.03 [b] | 5.84 ± 0.06 [b] | 5.63 ± 0.11 [b] |
| Acetic acid (g/L) | <0.03 | <0.03 | <0.03 | <0.03 |
| Lactic acid (g/L) | <0.02 | <0.02 | <0.02 | <0.02 |
| Malic acid (g/L) | 0.95 ± 0.03 [a] | 1.20 ± 0.04 [b] | 1.550 ± 0.05 [b] | 2.10 ± 0.02 [c] |
| Citric acid (mg/L) | 116.53 ± 8.73 [a] | 182.41 ± 12.53 [a] | 250.32 ± 29.51 [a] | 638.15 ± 72.51 [b] |
| End of fermentation (t = 8 days) | | | | |
| pH | 4.06 ± 0.02 [a] | 3.92 ± 0.05 [b] | 3.93 ± 0.01 [b] | 3.66 ± 0.01 [c] |
| Acetic acid (g/L) | 0.18 ± 0.03 [a] | 0.49 ± 0.07 [b] | 0.62 ± 0.04 [bc] | 0.80 ± 0.08 [c] |
| Lactic acid (g/L) | 1.70 ± 0.20 [a] | 2.55 ± 0.191 [b] | 3.22 ± 0.18 [bc] | 3.67 ± 0.18 [c] |
| Malic acid (g/L) | <0.03 | <0.03 | <0.03 | <0.03 |
| Citric acid (mg/L) | <11.00 | <11.00 | <11.00 | <11.00 |
| *Fermentation Quotient (FQ)* | 6.48 ± 0.79 [a] | 3.53 ± 0.20 [b] | 3.47 ± 0.23 [b] | 3.06 ± 0.17 [b] |

Values are mean ± standard deviation of triplicates. Significant differences between samples are indicated by different superscript letters ($p < 0.05$) for each compound.

With reference to acids at the end of fermentation (8 days), the addition of 0.5% Cava lees had a significant effect on acetic acid and L-lactic acid, with a major change occurring with the addition of 2%, compared to the control. D-lactic acid presented significance when lees were added at a minimum of 1% (*w/w*), compared to the control. Following the same tendency as wheat sourdoughs, rye fermentations also raised their organic acid production accordingly, with greater cell density and a lower pH, with the addition of Cava lees to their formulation.

Concerning FQ, by adding 0.5% (*w/w*) Cava lees, it was reduced to half, but there were no significant differences with the addition of higher concentrations of lees.

As previously stated, the microbial association due to their metabolism preferences is very common. As a consequence, the nutrient consumption affects the production of organic acids such as acetic and lactic acids, since yeasts may consume the soluble carbohydrates faster, decreasing LAB acidification because of the microbial competition [37].

Additionally, LAB metabolize malic acid and convert it to lactic acid [38]. In both wheat and rye sourdoughs, L-malic acid increases its concentration when Cava lees are added (Table 2); therefore, Cava lees may be a source of malic acid that might have been adsorbed during Cava ageing. Furthermore, some LAB strains are able to degrade tartrate, a major compound found in wine lees, into lactate and acetate [38]. Therefore, this could explain the increment in lactic acid concentration in sourdoughs with lees.

Citric acid metabolism can also produce acetate (considered an antimicrobial compound), as well as acetoin, diacetyl and butanediol. These compounds are flavor com-

pounds of the bread crumb [39,40]. Hence, a greater concentration of citric acid at the beginning of fermentation may result in an increased amount of these compounds, probably having a positive impact on flavor.

Considering the variables studied, the results obtained were subjected to a PCA (Figure 4) that confirmed the differences between the sourdoughs with and without Cava lees. The PCA shows similar behavior in sourdoughs with the same Cava lees concentration in both wheat and rye, although it can differ across samples according to the flour used to produce them. It can be observed that Component 1 separates the samples according to the percentage of lees added, whereas Component 2, which is equivalent to 22% of the variance, separates the sourdoughs according to the type of flour, wheat or rye. In fact, it shows that sourdoughs with Cava lees have higher LAB cell densities, acetic, citric and lactic acids as well as a lower pH, whereas control sourdoughs are defined with higher pH values, less microbial cell density and lower organic acids concentrations, especially wheat control sourdoughs. Therefore, according to the PCA, the addition of 2% (*w/w*) Cava lees to sourdough formulation has the greatest impact in its microbial populations and, consequently, to its physicochemical characteristics.

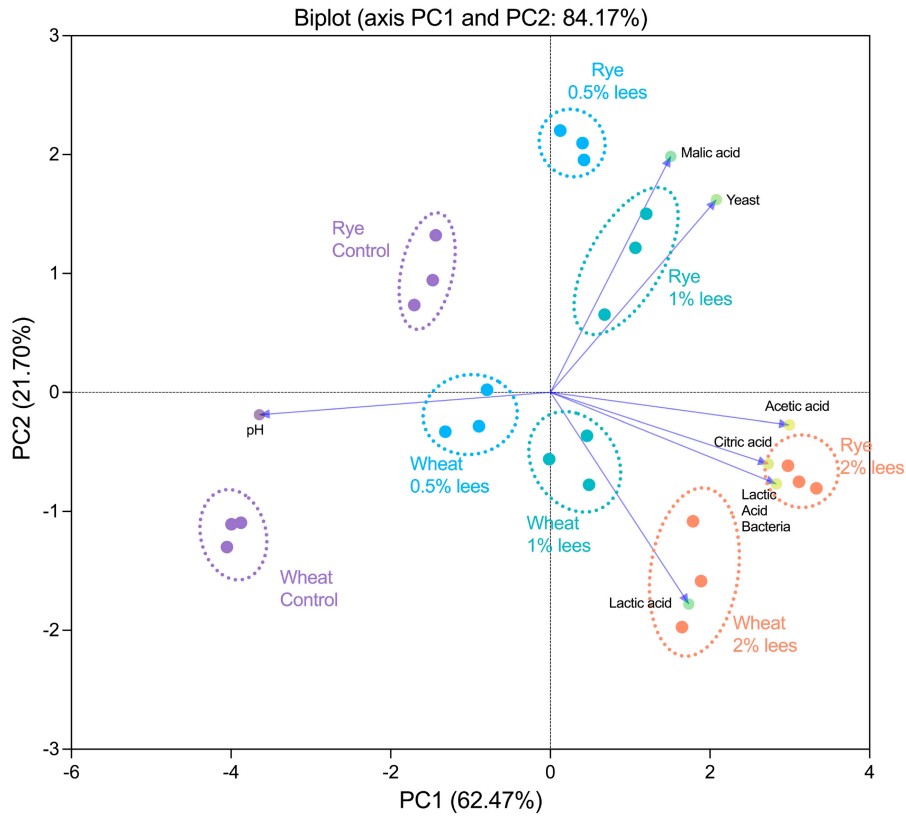

**Figure 4.** Principal component analysis (PCA) biplot of sourdoughs obtained at the end of fermentation.

## 4. Conclusions

Cava lees are a wine industry by-product containing several highly valuable compounds such as β-glucans and mannoproteins [1,4], with the potential to modify food-fermenting microbial populations, such as the ones in sourdough.

In this study, with the aim of revalorizing such by-products, it was found that the addition of 2% (*w/w*) Cava lees to sourdough formulation improved the growth and survival of LAB and yeasts that carry fermentation, especially in rye sourdough.

In addition, with increased microorganism cell density, there can be a greater production of organic acids and a lower pH, as shown in the PCA. Therefore, it may change sourdough bread flavor as well as other parameters such as texture and shelf life.

Since consumer acceptance is of great value, studies on sourdough bread volatiles and more complete sensory analysis should be conducted. Further studies with higher concentrations of Cava lees should also be considered, as well as the use of lees obtained from different Cava productions (e.g., different ageing times or initial coupages), which could also affect their composition and, consequently, bread flavor.

**Author Contributions:** Conceptualization, E.L.-T.; investigation, A.M.-G.; writing—original draft preparation, A.M.-G.; writing—review and editing, M.R.-A. and E.L.-T.; supervision, M.R.-A.; project administration, E.L.-T. All authors have read and agreed to the published version of the manuscript.

**Funding:** This research was funded by Comisión Interministerial de Ciencia y Tecnología (CICYT) (Spain) AGL2016-78324-R; the Generalitat de Catalunya, Project 2017-1376 SGR; INSA-UB (Institut de Recerca en Nutrició i Seguretat Alimentària), by XIA (Xarxa d'Innovació Alimentària); and Chartier World Lab through a grant from the Gouvernement du Québec to PhD student Alba Martín-Garcia.

**Institutional Review Board Statement:** Not applicable.

**Informed Consent Statement:** Not applicable.

**Data Availability Statement:** Not applicable.

**Acknowledgments:** The authors thank Freixenet S.A. for providing the Cava lees used in this study.

**Conflicts of Interest:** The authors declare no conflict of interest.

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
