# Peer review of "Revalorization of Cava (Spanish Sparkling Wine) Lees on Sourdough Fermentation"

_fermentation, doi:10.3390/fermentation8030133_

Round 1

Reviewer 1 Report

Dear authors, I have enjoyed reading your manuscript. I found interesting the use of sub-products from the food industry in other industrial process. I just had the feeling that the whole experimental results were split in two and this one part seems somehow short. The proposed further studies including volatile profile and sensory evaluation could’ve perfectly fit in this manuscript as well in order to complete the analysis.

I would like to address the following comments and suggestions:

Line 76: where do these lees come from? Were they prepared in liquid media, were they obtained from Cava production? Where the lees washed as to remove products/compounds other than the oligosaccharides?

Line 86: “were added to”

Line 216: with 1% or 2%?

Lines 219-222: could the yeast also be responsible of the increment in acetic acid production?

Lines 254-255: No need to rewrite the information again. Refer the difference in sugar composition in a shorter phrase.

Line 269: is the impact in flavor expected to be positive or not? Or, how would it impact the flavor?

Author Response

Thank you very much for your interest in the paper. Also, thank you for your comments and suggestions, they are correct and we have taken them into account. We agree that adding volatile composition and sensory characteristics could be very interesting. Therefore, we are working on the data of this part that we plan to present in another publication. We also thought that it would be too long to include all the parameters in a unique paper.

Line 76: where do these lees come from? Were they prepared in liquid media, were they obtained from Cava production? Where the lees washed as to remove products/compounds other than the oligosaccharides?

Response: They come from the winery of Freixenet S.A., they were obtained from cava production. The lees were centrifuged and lyophilised in order to reuse in the sourdough. We followed the method of Hernandez, et al., 2021. We add the explanation in the text.

Line 86: “were added to”

Response: it was corrected in the text.

Line 216: with 1% or 2%?

Response: it was 2% and it was corrected in the text.

Lines 219-222: could the yeast also be responsible of the increment in acetic acid production?

Response: Yes, according to the literature yeast could also produce acetic acid, but the concentration produced will be lower than the concentration produced by LAB. Also, we found higher plate counts of LAB than yeast. The development of yeast was added as Figure 2. The numbers of the rest of the figures have been updated.

Lines 254-255: No need to rewrite the information again. Refer the difference in sugar composition in a shorter phrase.

Response: we removed the sentence as the information could be found previously (lines 167-168).

Line 269: is the impact in flavor expected to be positive or not? Or, how would it impact the flavor?

Response: we clarify in the text. Probably, compounds as acetoin, butanediol and diacetyl will have a positive impact in the flavour as they have bakery and buttery notes and they are usual in bread flavour.

Reviewer 2 Report

in the keywords insert "prebiotic"

in the first column of table 1 replace flour rice / wheat flour

in results and discussion, in the comment to figure 1a explain why with 0.5% lees the LAB value is 8.1 log CFU / ml on the third day and decreases in the following days. (cells are dead? do you have the OD value?)

in results and discussion figure 1, it would be preferable to measure the increase in cell development considering that the values ​​at t0 are different.

in results and discussion, in the growth of yeasts why not make a figure with development data?

the figure with the growth of yeasts from t0 to t8 adds useful information to understand the growth of LAB.

in results and discussion lines 166 to 184, the comment is correct but without the yeast growth data it is only a hypothesis. some claims need to be supported by chemical analysis data.

in results and discussion paragraph 3.2.1. the first two lines are superfluous.

in results and discussion paragraph 3.2.2. the last lines "therefore, according to the PCA ......." it is better to insert them in the conclusions

Author Response

Thank you for your comments and suggestions, we will be taking them into account.

Prebiotic was added as a Keyword as the reviewer suggest. 

In the first column of table 1 replace flour rice / wheat flour.

Response: we added this information in the footnote.

In results and discussion, in the comment to figure 1a explain why with 0.5% lees the LAB value is 8.1 log CFU / ml on the third day and decreases in the following days. (cells are dead? do you have the OD value?)

Response: we didn’t measure the OD, It would have been interesting to have this information. It could be seen in the Figure as in the third day the value of log CFU/ml decrease, but it wasn’t significant. And then it was stable during time. Probably this decrease was an analytic question.

In results and discussion figure 1, it would be preferable to measure the increase in cell development considering that the values ​​at t0 are different.

Response: it could be interesting to measure the increase in cell development, but we are interested in the cell values at the end of fermentation.

In results and discussion, in the growth of yeasts why not make a figure with development data? the figure with the growth of yeasts from t0 to t8 adds useful information to understand the growth of LAB.

Response: thank for the suggestion, the figure about the development of yeast was added as Figure 2. The numbers of the rest of the figures have been updated.

 In results and discussion lines 166 to 184, the comment is correct but without the yeast growth data it is only a hypothesis. some claims need to be supported by chemical analysis data.

Response: with the new figure added (now Figure 2A and 2B) the comment was clarified.

In results and discussion paragraph 3.2.1. the first two lines are superfluous.

Response: the first lines of this paragraph are useful in order to discuss the pH values.

 In results and discussion paragraph 3.2.2. the last lines "therefore, according to the PCA ......." it is better to insert them in the conclusions

Response: the reviewer is correct; the sentence was rewritten. It is already explained in the conclusions section.